# Artificial Intelligence Applied to Colonoscopy: Is It Time to Take a Step Forward?

**DOI:** 10.3390/cancers15082193

**Published:** 2023-04-07

**Authors:** Antonio Z. Gimeno-García, Anjara Hernández-Pérez, David Nicolás-Pérez, Manuel Hernández-Guerra

**Affiliations:** 1Gastroenterology Department, Hospital Universitario de Canarias, 38200 San Cristóbal de La Laguna, Tenerife, Spain; 2Instituto Universitario de Tecnologías Biomédicas (ITB) & Centro de Investigación Biomédica de Canarias (CIBICAN), Internal Medicine Department, Universidad de La Laguna, 38200 San Cristóbal de La Laguna, Tenerife, Spain

**Keywords:** colonoscopy, artificial intelligence, CADe, CADx

## Abstract

**Simple Summary:**

In recent years, there has been an exponential rise in artificial intelligence-based technology. Artificial intelligence has been applied to several medical disciplines, such as gastroenterology. In the field of endoscopy, a wide variety of applications for artificial intelligence algorithms have been developed or are in a process of improvement. Computer-aided polyp detection and characterization are two of the most studied applications. In addition, there are several reports of other potential applications, such as the assessment of bowel preparation quality, while another future prospect is the prediction of cancer invasion depth. However, certain concerns remain, such as the universal use of this technology in clinical practice, impact on the incidence of interval colorectal cancer, cost-effectiveness, workload and patient burden.

**Abstract:**

Growing evidence indicates that artificial intelligence (AI) applied to medicine is here to stay. In gastroenterology, AI computer vision applications have been stated as a research priority. The two main AI system categories are computer-aided polyp detection (CADe) and computer-assisted diagnosis (CADx). However, other fields of expansion are those related to colonoscopy quality, such as methods to objectively assess colon cleansing during the colonoscopy, as well as devices to automatically predict and improve bowel cleansing before the examination, predict deep submucosal invasion, obtain a reliable measurement of colorectal polyps and accurately locate colorectal lesions in the colon. Although growing evidence indicates that AI systems could improve some of these quality metrics, there are concerns regarding cost-effectiveness, and large and multicentric randomized studies with strong outcomes, such as post-colonoscopy colorectal cancer incidence and mortality, are lacking. The integration of all these tasks into one quality-improvement device could facilitate the incorporation of AI systems in clinical practice. In this manuscript, the current status of the role of AI in colonoscopy is reviewed, as well as its current applications, drawbacks and areas for improvement.

## 1. Introduction

Colonoscopy is the gold standard for the detection and treatment of colorectal neoplastic lesions and is the cornerstone examination for colorectal cancer (CRC) screening [1]. Quality criteria, such as cecal intubation, withdrawal time, adequate photodocumentation and treatment of the lesions and bowel cleansing, have been proposed to improve the efficiency of the technique [2]. The adenoma detection rate (ADR) is deemed a surrogate marker of incidental CRC. Although US and European guidelines recommend an ADR of 20–30% in screening colonoscopy and 35–45% in fecal occult blood test-based CRC screening colonoscopy [3,4,5], the percentage of missing lesions can reach up to 27% [6], and the variability in the ADR among endoscopists is high. Indeed, the efficacy of the colonoscopy largely depends on the endoscopist’s skills, such as mucosal exposure during the examination, skills for lesion detection, diagnosis and proper removal. In addition, fatigue or distraction are factors that could decrease the performance of endoscopists [7]. Currently, we are witnessing a breakthrough in the development of artificial intelligence (AI)-based applications in medicine and in the field of gastroenterology, particularly in gastrointestinal endoscopy [8]. Recently, the American Association of Gastrointestinal Endoscopy (ASGE) and European Society of Gastrointestinal Endoscopy (ESGE) have established a priority guide for AI applied to digestive endoscopy [9,10]. Priority goals for development include vision-enhancing applications that place computer-aided polyp detection (CADe) and computer-aided polyp diagnosis (CADx) as one of the most necessary priorities, given the importance and workload of CRC screening and postpolypectomy surveillance. Other fields of research with important implications in clinical practice are the prediction of the depth of submucosal invasion in early colorectal cancer, cleansing quality issues such as a more objective assessment of bowel preparation during colonoscopy and the prediction of bowel cleansing before colonoscopy. Other tasks related to the lesion, such as location, size or even the assessment of the postpolypectomy resection site just after the polypectomy or in surveillance colonoscopies, are also aims of research. All these tasks are of paramount importance, and they have been included in quality colonoscopy guidelines [3]. Although AI applied to colonoscopy has the potential to improve quality metrics, resulting in more standardized quality indicators and increasing the performance of endoscopists, potential drawbacks such as cost-effectiveness, false-positive detections or deskilling should also be addressed. The aim of this review is to analyze the current evidence of AI applications in the colonoscopy setting and their current state of development.

## 2. Computer-Aided Polyp Detection (CADe)

Post-colonoscopy CRC (PCCRC) is the most important quality parameter in colonoscopy [11]. The incidence of diagnosed CRC has been reported to be between 3.6% and 9.3% [12]. Although various factors may be involved in the finding of PCCRC, the missing of advanced neoplastic lesions (advanced adenomas or serrated polyps) is a major issue [6]. In a recent study carried out in the UK, 58% of PCCRCs were attributed to quality deficiencies of the endoscopic procedure [13]. Another study suggested that the missing of neoplastic lesions after a proper or inadequate examination of the colon was the most plausible explanation, accounting for 91% of PCCRC [11]. ADR is a colonoscopy quality parameter closely associated with the PPCRC. It has been reported that for each 1% increase in ADR, the PPCRC incidence and mortality decrease by 3% and 5%, respectively [14]. However, interobserver variability in ADR is high, ranging from 7.4% to 52.5% in one study [14]. AI can be of benefit in this setting, reducing inter- and intraobserver variability and achieving the ADR benchmarks proposed by scientific societies regardless of the level of expertise or other constraints. CADe devices combine different machine-learning and deep learning methods. These systems are able to alert the endoscopist of potential lesions in real time with visual prompts and have the potential to decrease the polyp miss rate, contributing to improving ADR [15,16]. Indeed, several randomized controlled trials (RCTs) have shown that CADe increases the ADR and polyp detection rate (PDR) compared to conventional colonoscopy (CC) [17,18]. Table 1 shows the meta-analyses of randomized controlled trials (RCTs) [19,20,21,22,23,24,25,26,27,28] that assessed the benefit of CADe in ADR as the primary outcome. All these meta-analyses agree that ADR and PDR are higher using CADe compared with CC.

CADe can detect previously unrecognized lesions and serve as “a second set of eyes” by continuously monitoring mechanisms. Furthermore, these systems may help to reduce in-day variations in ADRs due to operator fatigue [21]. However, although adenoma detection is improving by use of CADe, there are some concerns about the real benefit of preventing CRC and cost-effectiveness. Indeed, the major benefit is at the expense of small adenomas, mainly diminutive adenomas. This is not surprising; firstly, endoscopists are more likely to miss small polyps rather than large ones, and secondly, it only represents the real life situation since diminutive polyps (less than 5 mm in size) comprise the great majority of polyps found in daily clinical practice [29]. From this perspective, it may be conceived as a quality marker if “even the smallest lesions are detected”. Conversely, it can also be argued that advanced adenomas are the real surrogate markers of CRC; they are usually larger and easy to detect by the human eye. This suggests that implementing CADe in clinical practice may not lead to a significant reduction in PCCRC. As a result, the current guidelines should be re-evaluated regarding the role of ADR as a surrogate marker of CRC. Although no metanalysis found an increased rate of advanced-adenoma detection, some of them found a higher ADR or adenomas per colonoscopy larger than 10 mm in size using CADe [19]. Another important issue, currently in discussion, is the benefit of CADe in the detection of serrated lesions. These lesions are sometimes difficult to recognize during colonoscopy due to their frequent flat morphology, imprecise margins and a color similar to that of the surrounding mucosa. It is hypothesized that these characteristics justify the strong association between the serrated pathway of carcinogenesis and interval CRC [30]. Five meta-analyses have shown outcomes regarding serrated lesions (Table 1) [22,23,25,27,28]. In three studies, statistically significant results were obtained [22,23,27]. Hassan et al. [22] and Huang et al. [23] observed significantly more serrated lesions per colonoscopy by using CADe. Shan et al. showed that there was a 78% reduction in the sessile serrated lesion miss rate with CADe. Therefore, AI seems to be of special interest for the detection of these serrated lesions.

Another concern is that CADe could necessitate a greater investment of time. It has been demonstrated that an increase in withdrawal time is associated with an increase in ADR [31]. Although, as shown in Table 1, CADe significantly increased the withdrawal time in some metanalyses [19,25]; when the biopsy time was excluded, this increase was not clinically significant (on average 0.3 to 0.46 min more in the CADe group).

The high sensitivity of CADe systems has been stated as a major drawback, particularly when a false-positive detection (FP) is defined as an area detected by the CADe systems of any duration during the withdrawal phase that was not deemed to be a real colorectal lesion by the reviewer [32]. The reported causes of FP are artifacts due to the mucosal wall, intestinal content, wrinkled mucosa, bubbles and local inflammation [32]. Hassan et al. [32] showed that the median number of FPs per colonoscopy was 27, and the overall time spent for each FP was negligible (0.2 ± 0.9 s). They concluded that 95% of FPs do not require additional time because they are easily eliminated as true lesions by endoscopists, and FP detections account for only 1% of the whole colonoscopy time. However, although FP may not be a relevant issue, it can cause fatigue in the endoscopist due to the necessity of a higher level of concentration to focus and rule out real lesions. A 2021 systematic review that included nine real-time video studies aimed to determine methods to minimize the drawbacks of FPs [33]. A frequent cause of a high number of FPs was less-than-ideal bowel preparation. This study proposed a water exchange technique (colonoscopy insertion with the infusion of warm water instead of air) to wash away feces and bubbles during insertion^,^ theoretically resulting in a lower percentage of FPs [34].

Other issues that arouse controversy are the impact of colon cleansing on CADe, the usefulness of CADe depending on endoscopist expertise and the risk of losing colonoscopy skills (deskilling) derived from the overreliance on CADe. Regarding the behavior of CADe depending on the different cleansing qualities, one recent RCT comparing CADe and CC in 370 outpatients found statistically significant differences in favor of the CADe group for the flat ADR in patients with adequate cleansing (39.8% vs. 24.4%; *p* = 0.03) [17]. However, there were no statistically significant differences in patients with excellent bowel cleansing when CADe and CC were compared. This study suggests that the better the cleansing quality is, the lower the difference between CADe and CC.

Some studies have suggested that expert endoscopists may benefit less from CADe [35,36]. However, Repici et al. [37] in a post hoc analysis pooling the data of two RCTs compared a CADe and a control group. In one of the studies, only expert endoscopists were included, while the other only involved nonexpert endoscopists (lifetime number of colonoscopies less than 2000). The assigned group (CADe or CC) was associated with the ADR but not the level of expertise. Similarly, an RCT performed by our group [17] stratified endoscopists into high detectors (HDs) and low detectors (LDs) by using an ADR cutoff of 40%, depending on the ADR in screening colonoscopies before the commencement of the study. They did not find any significant difference in ADR between both groups of endoscopists, and a nonsignificant trend was found in both HDs and LDs when CADe was used. Deskilling has been stated as a downside of AI in colonoscopy [38]. However, this has not been proven thus far and probably depends on the scenario in which CADe is used. If CADe is used as a “concurrent read” alongside the endoscopist, it could even increase the endoscopist’s skill to detect subtle lesions [18]. Conversely, if the endoscopist delegates all the responsibility of the detection in the CADe, there is probably a risk of deskilling. Hassan et al. [39] showed that CADe detected polyps faster than the human eye but also had a high false positive rate. It was postulated that this could cause the examiner to become overly reliant on the AI and potentially result in a loss of endoscopic skills, especially for non-expert endoscopists. A recent experimental study used colonoscopy videos with and without CADe to assess the reaction time, visual gaze pattern and the effect of false positive detections on the misinterpretation of mucosal surface as polyps [40]. This study showed that with the use of CADe, eyes traveled a shorter distance, and endoscopists falsely identified more polyps, regardless of their degree of expertise. The authors suggested that overreliance on CADe could lead to a potential risk of deskilling.

It is also worth discussing the benefit of CADe compared with other technical improvements aimed at increasing the ADR. These techniques could be classified as those that enhance the contrast between neoplastic and healthy mucosa, such as dye-spray chromoendoscopy or virtual chromoendoscopy, and systems that increase mucosal exposure (add-on devices). In the metanalysis of Spadaccini et al. the ADR was significantly higher with the use of CADe than with add-on devices (OR 1.54, 95% CI 1.22–1.94) and chromoendoscopy (OR 1.45, 95% CI 1.14–1.85). CADe was also a dominant strategy for the detection of sessile serrated lesions [41]. A recent RCT suggested that the combination of CADe plus add-on-devices could be better than each of them analyzed separately [42].

### 2.1. Computed Aid Quality Improvement (CAQ)

Computer-aided quality improvement (CAQ) for real-time monitoring of withdrawal speed has been tested to increase ADR, with good results [43]. Since the benefit of CADe systems depends on the amount of exposed mucosa, it would seem logical to assume that the combination of CADe and CAQ could increase the ADR. This hypothesis was tested in a recent randomized controlled study including 1076 patients assigned to a control group (regular practice), CADe, CAQ and the combination of CADe and CAQ [43]. ADR was significantly higher in the combination group than in the CADe and control groups. Although the ADR was also higher in the combination group than in the CAQ group, the difference was not statistically significant. However, to improve the ADR, it makes sense that this type of technology should cover all dimensions of colonoscopy quality. Some studies have shown that tiredness can decrease the ADR in such a way that endoscopists working the full day detect fewer adenomas in afternoon colonoscopies [44]. A recent study aimed to assess whether the use of AI systems (CADe, CAQ or the combination of both) may overcome the tiredness effect [45]. A total of 1780 patients enrolled in two randomized controlled studies that compared AI systems with conventional colonoscopy were included. While in the conventional colonoscopy group, the ADR declined with each hourly interval (13.73% vs. 5.70%; *p* = 0.005; OR, 2.42; 95% CI, 1.31–4.47), it was not a significant factor in the AI system group (22.95% vs. 22.06%, *p* = 0.78; OR, 0.96; 95% CI; 0.71–1.29).

### 2.2. Computer-Aided Polyp Diagnosis (CADx)

Management of polyps is based on a precise optical diagnosis to choose the proper resection technique, either by endoscopy or surgery, and then carry out the subsequent histological analysis. The Preservation and Incorporation of Valuable endoscopic Innovations (PIVI) initiative was developed to assist the colonoscopic management of diminutive polyps located in the rectosigmoid colon based on the limited clinical importance of these polyps and with the main purpose of decreasing costs [46]. However, even with the assistance of enhanced imaging techniques such as narrow band imaging (NBI), PIVI thresholds for resecting and discarding that prevent histological analysis of tiny hyperplastic lesions (90% agreement in the assignment of postpolypectomy surveillance intervals between histology and optical diagnosis) and leave in situ strategy and that give a recommendation for postpolypectomy endoscopic surveillance just after the endoscopic procedure (90% negative predictive value for adenomatous histology) are only met in expert centers [4].

One of the potential applications of AI in colonoscopy is to provide an accurate, almost instantaneous prediction of whether a polyp is neoplastic. With a high rate of histological prediction through AI, endoscopists could adopt a resect and discard strategy, obviating the need for complementary histopathological studies in tiny polyps [47]. Some AI systems already exceed these thresholds in the experimental environment, combining CADx with NBI or endocytoscopy [48,49]. Appendix A shows several real-time studies that employed AI systems combining them with other advanced techniques, such as narrow band imaging, magnification, near-focus, blue light imaging (BLI), endocytoscopy, confocal endomicroscopy or laser-induced autofluorescence. These AI systems were trained with different types of information sources (still images and/or video images). They showed a negative predictive value for the detection of neoplastic lesions higher than the threshold value of 90% established for the diagnosis-and-leave criterion established by the ASGE [46,48,50,51,52,53,54].

However, caution should be exerted with the application of these AI systems in clinical practice given a lower diagnostic accuracy in the proximal colon compared with the sigmoid and rectum [49]. In addition, internal and external validation of algorithms should be mandatory. It is also important to know how the AI system (training set quality) has been trained, concerning the number and type of lesions included according to their morphology and size. The number and qualification of the observers who classify the images is not trivial either. Unfortunately, this information is not always provided in the reported studies or by the companies trading these systems (24). Despite the complex and scarcely spreading endoscopic diagnosis techniques used in some studies (i.e., endocytoscopy, blue light imaging, autofluorescence), others using white-light endoscopy (WLE) also showed values higher than the threshold established by ASGE [55,56,57].

The theoretical benefits of this strategy would be cost savings, a shorter procedure duration and fewer adverse effects by avoiding unnecessary polypectomies [46,58]. It has been suggested that these systems could decrease the learning curve in non-expert endoscopists. One study assessed whether a CNN trained for the characterization of colorectal polyps could increase the skills of endoscopists with different experience. The study found that the use of CNN led to an improvement in non-expert endoscopists (73.8% to 85.6%, *p* < 0.05) for colorectal polyp characterization, who almost reached the accuracy of experts (89.0%, *p* = 0.10) [59].

The implementation of hybrid AI systems, packed into an “all-in-one” solution that includes both detection of lesions (CADe) and their characterization (CADx), should enable a higher detection rate of adenomas and serrated adenomas, compensated by a precise selection of candidate lesions for resection. This would result in a reduction in the incidence of PCCRC at reasonable costs. A cost-effectiveness analysis using a Markov model with microsimulation, which compared colonoscopy with and without AI for colorectal cancer screening for individuals at average risk, estimated an incremental gain of 4.8% in the reduction of CRC incidence when colonoscopy vs. colonoscopy plus AI were compared (44.2 vs. 48.9%), with a savings per individual of $57 [60]. Hybrid AI systems may be especially relevant in contexts such as colonoscopies by nonexpert endoscopists with low ADR or in fecal occult blood test-based screening programs where the prevalence of neoplastic lesions is higher. However, there are concerns regarding the usefulness of certain endoscopic technological innovations in CRC screening programs since they do not always increase the ADR when patients undergo high-quality colonoscopy by endoscopists with a high ADR [51,52,59,61,62,63,64].

### 2.3. Prediction of the Depth of Submucosal Invasion

Enhanced imaging techniques have been used for predicting submucosal invasion. The international classification NICE (Narrow band imaging International Colorectal Endoscopic classification) [65] and JNET (Japanese Expert Team NBI classification) [66] rely on the examination of surface and vascular patterns for predicting deep submucosal invasion. Unfortunately, training for this prediction is not easy; the prediction is endoscopist dependent and requires experience, and, even in expert hands, the sensitivity for some patterns is suboptimal [67]. In two recent studies, the Type 2B pattern of the JNET classification only reached a sensitivity of 43–44% for differentiating high-grade dysplasia and shallow submucosal invasive carcinoma from deep submucosal invasion [67,68]. Overall, the accuracy for deep submucosal invasion ranged from 59% to 84% across the studies [69,70]. An inadequate diagnosis of deep submucosal invasion is relevant, leading to improper and harmful treatments, either due to overtreatment with the indication of surgery in patients with shallow neoplastic tumors (i.e., high-grade dysplastic tumors or with superficial submucosal invasion) or undertreatment, such as noncurative endoscopic resections with potential severe complications. AI may have the potential to overcome these issues and decrease interobserver variability.

There are no real-time trials in this setting thus far, and the published studies have aimed to develop and validate a convolutional neural network (CNN) to predict deep submucosal invasion. One limitation of these systems is that the prediction is usually based on still endoscopic images. Although some studies used images obtained with advanced technology such as NBI with magnification or endocytoscopy for training the CNN, this technology is not available in all endoscopy units. Appendix A shows recent studies addressing this task using WLE. In general, they are single-center studies, and the composition of training sets is variable. A higher performance is achieved when the studies include advanced CRCs; however, the performance is lower for differentiating noninvasive or shallow invasive early CRCs from deep invasive early CRCs. The studies showed that AI is superior to trainees for predicting deep submucosal invasion but not to experts’ assessment but seems to be faster [71,72,73]. In a recent multicenter study, the authors designed and validated a CNN that included a multimodal data analysis (clinical information plus WLE endoscopy images plus enhanced endoscopy images) along with prior knowledge (distal location, large size, Paris type, surface morphology and the Narrow Band Laser Imaging International Colorectal Endoscopic classification) [74]. The accuracy of this system for predicting deep submucosal invasion was 90.4%. This type of combined system seems to work better than the sole deep learning model.

### 2.4. Assessment of the Colon Preparation

Several studies have been carried out to train and validate CNNs for detecting bowel cleansing during colonoscopy based on validated cleansing scales [75]. These systems can also overcome the limitations of interobserver variability in rating colon cleansing during colonoscopy. ENDOANGEL is the only commercially available system for real-time assessment of colon cleansing [76]. This system was trained with still images and colonoscopy videos, achieving an accuracy for bowel preparation assessment between 93.3% and 89.4% and better than the performance of endoscopists [76]. ENDOANGEL was able to detect inadequate bowel cleansing in 100% of cases.

A recent study assessed in a randomized fashion the prediction of bowel cleansing before colonoscopy by a CNN trained with rectal effluents during bowel preparation [77]. This approach is especially interesting since it has the potential to guide rescuing strategies before the colonoscopy (i.e., recommending additional bowel preparation in those patients with a CNN prediction of inadequate cleansing). In this study, patients were randomized to real-time CNN assessment of the rectal effluent after bowel cleansing intake and the prediction reported by patients following a set of images resembling different qualities of rectal effluents. There was no significant difference between the two predictions. CNN prediction was compared with the Boston Bowel Preparation Scale (BBPS) during the colonoscopy; however, the capability of discrimination between adequate (BBPS ≥ 6 points) or inadequate (BBPS < 6 points) bowel preparation was poor, detecting only 8.5% of the patients who finally had inadequate bowel preparation during the colonoscopy. Despite this, this type of tool lays the foundation for research on additional cleansing strategies carried out before colonoscopy [77].

### 2.5. Other Issues

The estimation of polyp size is an important issue since it can impact the therapeutic approach as well as the recommended surveillance intervals. Endoscopic estimation cannot rely on visual estimation, and usually different tools, such as open snares or forceps, are used. However, this process is time-consuming and is not always exact. Measures after formalin fixation are also inaccurate because of shrinking and fragmentation of the sample [78]. A recent study aimed to assess the accuracy of an AI tool designed to measure polyp size. This measure relies on the distance between the main vessel branches. This approach proved to be a more accurate and reliable method of measurement than visual estimation or estimation assisted by open biopsy forceps during colonoscopy [79].

Another potential application of AI systems in colonoscopy is to accurately evaluate the location of the tip of the colonoscope during the examination. It is known that endoscopist assessment is unreliable [80]. A recent study trained a CNN with images of a magnetic endoscope imaging positioning device. Although the predicted location dividing the colon into five segments was suboptimal (overall accuracy of 63%, sensitivity of 63%, specificity of 89%), this study set the foundation for further investigation in this field [81].

## 3. Challenges, Drawbacks and Areas of Improvement

Currently, the implementation of AI in colonoscopy is not yet widespread. There are concerns regarding the training requirements of this technology, clinical benefits, proper indications and cost-effectiveness. Some issues remain regarding training and transparency of CNNs [82]. One critical factor for training a CNN is the quantity and quality of the training dataset. It must be large enough and representative of the data obtained under real conditions. In addition, images should be adequately labeled. Ideally, the software should be able to adapt to different and suboptimal conditions. Obtaining a representative dataset may be difficult due to the many variables to consider, such as differences in colonic anatomy, cleansing quality, colonic normality variants or endoscopist skills. In this scenario, class imbalance poses a challenge. This situation is a limitation for training a CNN under conditions of low prevalence. The prediction of the extent of invasion in early CRC would be an example.

Methods such as data augmentation (an artificial procedure for increasing the dataset) consisting of random rotations, vertical and horizontal flips and zoom can improve image classification. Additionally, active learning (an approach where an algorithm iteratively selects the most informative data samples to be labeled by a human expert) can make the CNN more robust.

Another important issue is transparency. It is crucial to evaluate commercial AI systems available before the implementation in clinical practice. These systems probably have different algorithms and datasets, as well as their own training and validation. It is of utmost importance to have access to the image features that were used to train the CNN. Local interpretation methods explain why the network predicted a particular image, while global interpretation methods explain the entire model as a whole, including its capabilities and limitations. There must be transparency regarding both local and global explanations. This information could be valuable for hospital management and endoscopists when deciding on the eventual implementation of these devices in clinical practice. Another area of improvement is the urgent need of a legal regulation for the use of AI in the healthcare setting [83]. Although several CADe and CADx systems are currently commercialized, there are concerns that this new technology could be a source of healthcare mistakes and breaches in patient privacy and data security. The lack of transparency often comes up in legal discussions about the use of AI. Liability is also a source of concern. Many researchers emphasize the lack of understanding of the functioning of AI, as it is like a black box that produces predictions but cannot explain its results [84]. Although there is currently no legal framework, some authors have suggested that the responsibility should be shared between the device constructors and the doctors who use it [83].

Computer-aided detection (CADe) may increase costs, at least in the short term, due to the number of biopsies, polyp resections and pathological assessments [60]. A potential concern is also the length of the procedure, due to a high false positive detection rate or the number of biopsies or polyp resections. However, these drawbacks could be mitigated by a reduction of CRC cancer incidence and mortality in the longer term, as shown in microsimulation modeling studies [60]. Ideally, long observational studies and randomized controlled trials should prove its clinical benefit. Deskilling, as mentioned above, is also a subject of concern. There is a risk of deskilling if endoscopists rely too heavily on the technology and do not continue to rely on their clinical skills and experience [40]. Although CADx has shown to meet PIVI requirements, it was not clearly better than optical diagnosis in prospective studies [85]. Therefore, in the era of optical diagnosis, its benefit could be limited to inexperienced endoscopists. Economic studies of CADx are warranted in different healthcare systems. Regarding the implementation of AI for predicting deep submucosal invasion, there is a lack of real-time trials, and the CNNs need some improvement to increase accuracy.

## 4. Conclusions and Future Directions

AI applied to gastrointestinal endoscopy has now come to stay. Currently, there are AI systems with different tasks and stages of development. Most of these systems have been developed to improve and standardize quality metrics in colonoscopy, especially surrogate markers such as the ADR and the number of adenomas per colonoscopy. Although, in general, most of these studies have shown a benefit for these tasks, multicenter, long-term studies with more robust endpoints, such as the impact on CRC incidence and mortality, are lacking. It should be noted that CADe has been shown to increase ADR, but at the expense of small and diminutive adenomas. Further research is needed to determine whether this translates into a reduction of CRC incidence and higher survival. Before the implementation of AI in clinical practice, some other issues require further research. First, the amount of colonic surface area examined is a crucial factor in determining colonoscopy quality. Although AI devices only provide information of lesions within the field of view, they could potentially help to improve mucosal exposure. This, in turn, could increase lesion detection rates regardless of CADe devices. This real-time information could improve the quality of the examination. However, currently, there is low evidence regarding what information should be provided, how it should be presented and where it should be shown. Initial studies with such systems demonstrated that the quality of colonoscopy could be similar to that performed by expert endoscopists [86]. Second, further efforts are needed to adapt AI tools to routine endoscopic technology. For instance, although recent studies have shown that AI-assisted polyp size measurement using laser technology can be more accurate than visual measurement, it requires special endoscopic equipment [87]. Another field of research is the integration of the AI throughout the colonoscopy process. The examination itself is just one part of the procedure, as endoscopists dedicate a significant amount of time after the colonoscopy to create the colonoscopy report, that includes quality measures such as withdrawal time and examined mucosal surface. AI can help to create structured reports and choose the most appropriate images, while also automatically rating cleansing quality based on validated cleansing scales. The integration of these AI systems in clinical practice can also provide individual and general data on the quality of endoscopists and endoscopy units, respectively. These data can be used for internal and external audits, which can help to meet quality standards and promote specific improvement strategies.

Reimbursement is a major issue and has been identified as an important barrier for the widespread implementation of AI in clinical practice. Indeed, only two studies have reported cost-effectiveness of CADe and CADx so far [60,88]. High quality cost-effectiveness analysis of these devices in different healthcare systems with different reimbursement policies is also needed before their implementation in routine clinical practice. Cost-effectiveness analyses of these systems are also needed before their implementation in routine clinical practice. The development of multifunctional AI systems capable of performing different tasks with the same device may increase the technical quality of the endoscopic procedure in all its dimensions and could address these challenges. However, once again, more complex economic studies should address the cost-effectiveness of these integrated systems.

Finally, a field of development is robotics. There are currently self-propelling colonoscopes available on the market [89]. However, it is possible that in the future, colonoscopy and the different tasks performed during the procedure could be autonomously guided by software.

## Figures and Tables

**Table 1 cancers-15-02193-t001:** Meta-analysis of randomized controlled trials using computer-aided polyp detection.

	Studies (N)	Patients (N, Range)	Indications	ADR *, CADe ** vs. CC ^†^ (OR/RR ^§^, 95% CI)	AADR ^‡^, CADe vs. CC (OR, 95% CI)	SSLDR ^#^, CADe vs. CC (OR, 95% CI)	APC ^¶^, CADe vs. CC (OR, 95% CI)
Shah S.J. 2022 [27]	14	10,928 (230–2352)	Symptoms and screening	1.52 (1.39–1.67), *p* = 0.04	NR	NR	≤10 mm: 2.29 (1.97–2.26), *p* < 0.001 >10 mm: 1.93 (1.18–3.16), *p* < 0.01
Huang D.2022 [23]	10	6629 (128–1058)	Screening and surveillance	1.45 (1.32–1.59), *p* < 0.001	NR	NR	1.66 (1.52–1.81), *p* < 0.001
Zhang Y. 2021 [28]	7	5427 (369–1058)	Symptoms and screening	1.72 (1.52–1.95), *p* < 0.001	0.70 (0.50–0.97), *p* = 0.03	0.87 (0.61–1.23), *p* = 0.43	NR
Nazarian S. 2021 [26]	8	5577 (150–1058)	Symptoms and screening	1.53 (1.32–1.77), *p* < 0.001	NR	NR	NR
Li J. 2021 [24]	5	4311 (623–1058)	Symptoms and screening	1.75 (1.52–2.01), *p* < 0.001	NR	NR	NR
Hassan C. 2021 [22]	5	4354 (623–1058)	Symptoms, screening and surveillance	1.44 (1.27–1.62), *p* < 0.01	1.35 (0.74–2.47), *p* = 0.33	NR	Overall: 1.70 (1.53–1.89), *p* < 0.01 ≤5 mm: 1.69 (1.48–1.84), *p* < 0.000 6–10 mm: 1.44 (1.19–1.75), *p* < 0.000 >10 mm: 1.46 (1.04–2.06), *p* < 0.03
Deliwala S. 2021 [21]	6	4996 (623–1058)	Symptoms and screening	1.77 (1.50–2.08), *p* < 0.001	NR	NR	NR
Barua I. 2021 [20]	5	4311 (623–1058)	Symptoms and screening	1.52 (1.31–1.77), *p* < 0.001	NR	NR	NR
Ashat M. 2021 [19]	6	5058 (659–1058)	Symptoms and screening	1.76 (1.55–2.00), *p* < 0.001	NR	NR	NR
Mohan B.P. 2021 [25]	6	4962 (623–1058)	Symptoms, screening and surveillance	1.50 (1.30–1.72), *p* < 0.0001	1.00 (0.74–1.36), *p* = 0.93	1.29 (0.89–1.89), *p* = 0.18	NR

Notes: * ADR: adenoma detection rate; ** CADe: computer-aided detection; ^†^ CC: conventional colonoscopy; ^‡^ AADR: advanced adenoma detection rate; ^#^ SSLPC: sessile serrated lesion detection per colonoscopy; **^¶^** APC: adenomas detection per colonoscopy; NR: non-reported; ^§^ OR: odds ratio, RR: relative risk. * All outcomes are statistically significant with a *p* value < 0.05, otherwise the *p* value is specified.

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
