# Peer review of "Artificial Intelligence Applied to Colonoscopy: Is It Time to Take a Step Forward?"

_cancers, 2023, doi:10.3390/cancers15082193_

Round 1

Reviewer 1 Report

To the authors of: "Artificial Intelligence applied...."

Congratulation on a very interresting and comprehensive review. The article is well written and mostly easy to understand. The topic is of high interrest and the references represented well ballanced.

I agree that evidence is accumulating to a degree where a descision could be made on a more widespread routine use of AI support in colonoscopy. I do not, however understand the arguments for collecting all algorithms in one package. 

Some conserns:

1.

The tables, especially table 1, are very difficult. The lay out with two sections (not explained), with poorly aligned rows and lines does reading difficult.

Should all 10 papers be aligned ? Is there a reason for this divition? In some cases more answers are given to one cell and some cells are emphty. Sometimes a p value is presented and sometimes not. I cant see how for example the SSLPC CADe vs CC kan be higher than 1 (1.53), and the SSLDR CADe vs CC is lower than 1 (0.87). The tables are simply too difficult. 

Table one should be improved and table 2 and 3 could perhaps be submitted as supplementary material.

2. 

The problems of using ADR as a pseudo endpoint is well described. When you describe the correlation between PDR and interval cancers, you should be clear on the fact that the added PDR from CADe is very likely not representative for PDR overall. You show that PDR for large adenomas is not significantly improved, ad therefore we cannot use PDR as a surrugate in this context.

You state that the overrepresentation of diminuitive polyps merely reflects the size distribution in the polyp population. I disagree on that - its more likely because the endoscopist is more likely to miss small rather than large polyps. And this would change the arguments  for CADe addition. Assuming that dininuitive and possible small polys does not inflict patient outcome significantly, the argument for CADe is then  gone. This conclusion is supported by other authors in resent publications (Barua I ).

Minor conserns: 

1- Page 7 line 203-4: Do not understand?

2. Page 10 line 233 ff: Internal and external validation of algorithms should be mandatory

3. Page 10 line 244 ff: Is the truth that filtering polyps/adenomas without significance is becoming a necessity to prevent poor cost bennefit, higher complication rates and overtreatment.

4. Page 10 line 252 ff: Discuss only significant adenomas.

Author Response

We appreciate the comments of the Reviewer. We sincerely think that the comments have contributes to improve the manuscript.

Reviewer 1

To the authors of: "Artificial Intelligence applied...."

Congratulation on a very interesting and comprehensive review. The article is well written and mostly easy to understand. The topic is of high interest and the references represented well balanced.

I agree that evidence is accumulating to a degree where a descision could be made on a more widespread routine use of AI support in colonoscopy. I do not, however understand the arguments for collecting all algorithms in one package. 

Some concerns:

1.The tables, especially table 1, are very difficult. The lay out with two sections (not explained), with poorly aligned rows and lines does reading difficult.

Should all 10 papers be aligned ? Is there a reason for this divition? In some cases more answers are given to one cell and some cells are emphty. Sometimes a p value is presented and sometimes not. I cant see how for example the SSLPC CADe vs CC kan be higher than 1 (1.53), and the SSLDR CADe vs CC is lower than 1 (0.87). The tables are simply too difficult. 

Table one should be improved and table 2 and 3 could perhaps be submitted as supplementary material.

Reply: Table 1 has been improved according with the reviewer. References in rows and variables in columns. We have also simplified table 1. Tables 2 and 3 have been submitted as supplemental material. A recent reference has been added to table 2 (Rondonotti E. Endoscopy 2023; 55: 14-22)

  1. The problems of using ADR as a pseudo endpoint is well described. When you describe the correlation between PDR and interval cancers, you should be clear on the fact that the added PDR from CADe is very likely not representative for PDR overall. You show that PDR for large adenomas is not significantly improved, ad therefore we cannot use PDR as a surrugate in this context.

Reply: We appreciate the comment of the reviewer. Certainty with the implementation of artificial intelligence in clinical practice, surrogate markers such as adenoma detection rate (ADR) are less important and guidelines should reflect this new reality in the current guidelines. A paragraph has been added in the text (Page 4, lines 114-117): “This suggests that implementing CADe in clinical practice may not lead to a significant reduction in PCCRC. As a result, the current guidelines should be re-evaluated regarding the role of ADR as a surrogate marker of CRC.”

You state that the overrepresentation of diminuitive polyps merely reflects the size distribution in the polyp population. I disagree on that - its more likely because the endoscopist is more likely to miss small rather than large polyps. And this would change the arguments  for CADe addition. Assuming that dininuitive and possible small polys does not inflict patient outcome significantly, the argument for CADe is then  gone. This conclusion is supported by other authors in resent publications (Barua I ).

Reply: We thank the reviewer for this comment. We agree with the reviewer that both factors may influence the high rate of diminutive adenomas detected by CADe. A sentence addressing the reviewer´s observation has been added to the manuscript (Page 4 line 108-109): “Firstly, endoscopists are more likely to miss small polyps rather than large ones”

Minor concerns: 

1.Page 7 line 203-4: Do not understand?

Reply: Sorry. In the next paragraph we will try to better explain the purpose of the Preservation and Incorporation of Valuable Endoscopic Innovations for diminutive colorectal polyps. One of the potential applications and goal of artificial intelligence in colonoscopy is to provide an accurate, almost instantaneous prediction of whether a polyp is precancerous or non-precancerous. This information allows the endoscopist to decide if and how to remove a colonic polyp. With a high capacity for histological prediction through artificial intelligence, the endoscopist could adopt a resect and discard strategy, obviating the need for complementary histopathological studies in hyperplastic tiny polyps (less than 5 mm).In this regard, the American Society of Gastrointestinal Endoscopy has proposed that: 1) for a technology to be used to guide the decision to leave suspected rectosigmoid hyperplastic polyps 5 mm or smaller in place (without resection), the technology should provide a 90% or greater negative predictive value (NPV) (when used with high confidence) for adenomatous histology, and 2) for colorectal polyps 5 mm or smaller to be resected and discarded without pathologic assessment, endoscopic technology (when used with high confidence) used to determine histology of these polyps, should provide 90% or greater agreement in assignment of postpolypectomy surveillance intervals when compared with decisions based on pathology assessment of all identified polyps (Rex DK, Kahi C, O'Brien M et al. The American Society for Gastrointestinal Endoscopy PIVI (Preservation and Incorporation of Valuable Endoscopic Innovations) on real-time endoscopic assessment of the histology of  diminutive colorectal polyps. Gastrointest Endosc 2011; 73: 419-422; Rex DK, et al. Gastrointest Endosc 2015; 81: 502 e501-502 e516).

In the new version of the manuscript the paragraph has been modified (page 6, lines 224-231): "However, even with the assistance of enhanced imaging techniques such as narrow band imaging (NBI), PIVI thresholds for resecting and discarding strategy, that prevents histological analysis of tiny hyperplastic lesions (90% agreement in the assignment of postpolypectomy surveillance intervals between histology and optical diagnosis) and leave in situ strategy, that gives a recommendation for postpolypectomy endoscopic surveillance just after the endoscopic procedure (90% negative predictive value for adenomatous histology)  are only met in expert centers."

  1. Page 10 line 233 ff: Internal and external validation of algorithms should be mandatory

Reply: This sentence has been added in the new version of the manuscript (Page 6, lines 249-250): “In addition, internal and external validation of algorithms should be mandatory.”

  1. Page 10 line 244 ff: Is the truth that filtering polyps/adenomas without significance is becoming a necessity to prevent poor cost bennefit, higher complication rates and overtreatment.

Reply: Thank you very much for this comment. A recent study demonstrated that the resect and discard strategy offers substantial cost savings (Orlovic M. Gastrointest Endosc 2023. In press.doi: 10.1016/j.gie.2023.01.054). It is also known that diminutive and small hyperplastic polyps specially in the rectum and sigmoid colon are not a threat, and thereby treatment of these lesions are not justified (Rex DK. Gastrointestinal endoscopy 2011; 73: 419-422. doi: 10.1016/j.gie.2011.01.023. These references are included in page 6, line 259.

  1. Page 10 line 252 ff: Discuss only significant adenomas.

Reply: The  study of Areia et al., (Lancet Digit Health 2022; 4: e436-e444), found in a simulation model, that AI during screening colonoscopy might be cost-effective to reduce incidence and mortality of colorectal cancer and also that “Screening colonoscopy reduced colorectal cancer mortality from 2393 (2·4%) deaths per 100 000 people to 1227 (1·2%) deaths per 100 000 due to both colorectal cancer prevention owing to increased adenoma detection and removal, and diagnosis of colorectal cancer at earlier stages with consequent improved survival.” As shown in table 3 (original from that study) more high-risk adenomas and low-risk adenomas were diagnosed with the artificial intelligence. 

Reviewer 2 Report

·       There is not much discussion about the drawbacks and areas for improvement. Also, Section 3. Conclusions and future directions, does not include enough future direction.

·       The following papers need to be considered to be added:

o   Lam, A. Y., Duloy, A. M., & Keswani, R. N. (2022). Quality Indicators for the Detection and Removal of Colorectal Polyps and Interventions to Improve Them. Gastrointestinal Endoscopy Clinics, 32(2), 329-349.

o   Gubbiotti, A., Spadaccini, M., Badalamenti, M., Hassan, C., & Repici, A. (2022). Key factors for improving adenoma detection rate. Expert Review of Gastroenterology & Hepatology, 16(9), 819-833.

o   Ahmad, A., Wilson, A., Haycock, A., Humphries, A., Monahan, K., Suzuki, N., ... & Saunders, B. P. (2022). Evaluation of a real-time computer-aided polyp detection system during screening colonoscopy: AI-DETECT study. Endoscopy.

o   Troya, J., Fitting, D., Brand, M., Sudarevic, B., Kather, J. N., Meining, A., & Hann, A. (2022). The influence of computer-aided polyp detection systems on reaction time for polyp detection and eye gaze. Endoscopy, (AAM).

o   Brown, J.R.G., Mansour, N.M., Wang, P., Chuchuca, M.A., Minchenberg, S.B., Chandnani, M., Liu, L., Gross, S.A., Sengupta, N. and Berzin, T.M., 2022. Deep learning computer-aided polyp detection reduces adenoma miss rate: a United States multi-center randomized tandem colonoscopy study (CADeT-CS Trial). Clinical Gastroenterology and Hepatology, 20(7), pp.1499-1507.

·       The authors need to explain more about Deskilling and its influence in different scenarios of polyps and respective changes in ADR due to deskilling.

Author Response

We appreciate the comments of the Reviewer. We sincerely think that the comments have contributes to improve the manuscript.

Reviewer 2

There is not much discussion about the drawbacks and areas for improvement.

Reply: we want to than the reviewer for this suggestion. A new paragraph has been included (please see point 3, 7 “Drawbacks and areas of improvement”, Page 8, lines 355-366 and Page 9, lines 367-372): “Currently, the implementation of AI in colonoscopy is not yet widespread. There are concerns regarding the clinical benefits, proper indications, cost-effectiveness, and training requirements of this technology. Computer-aided detection (CADe) may increase costs, at least in the short term, due to the number of biopsies, polyp resections, and pathological assessments (Areia, et al. Lancet Digit Health 2022;4:e436-44). A potential concern is also the length of the procedure, due to a high false positive detection rate or the number of biopsies or polyp resections. However, these drawbacks could be mitigated by a reduction of CRC cancer incidence and mortality in the longer term, as shown in microsimulation modeling studies (Areia, et al. Lancet Digit Health 2022;4:e436-44). Ideally, long observational studies and randomized controlled trials should prove its clinical benefit. Deskilling, as mentioned above, is also a subject of concern. There is a risk of deskilling if endoscopists rely too heavily on the technology and do not continue to rely on their clinical skills and experience(Troya, et al. Endoscopy 2022; 54:1009-14). Although, CADx has shown to meet PIVI requirements, it was not clearly better than optical diagnosis in prospective studies (Rondonotti E, et al. Endoscopy 2023; 55: 14-22). Therefore, in the era of optical diagnosis, its benefit could be limited to inexperienced endoscopists. Economic studies of CADx are warranted in different healthcare systems. Regarding the implementation of AI for predicting deep submucosal invasion, there is a lack of real-time trials, and the CNNs need some improvement to increase accuracy.”

Also, Section 3. Conclusions and future directions, does not include enough future direction.

Reply: Additional information has been included in this section (Page 9, lines 380-384): “ Reimbursement is a major issue and has been identified as an important barrier for the widespread implementation of AI in clinical practice. Indeed, only two studies have reported cost-effectiveness of CADe and CADx so far. High quality cost-effectiveness analysis of these devices in different healthcare systems with different reimbursement policies are also needed before their implementation in routine clinical practice” and Page 9 , lines 388-390: “However, once again, more complex economic studies should address the cost-effectiveness of these integrated systems.”

The following papers need to be considered to be added:

o   Lam, A. Y., Duloy, A. M., & Keswani, R. N. (2022). Quality Indicators for the Detection and Removal of Colorectal Polyps and Interventions to Improve Them. Gastrointestinal Endoscopy Clinics, 32(2), 329-349.

o   Gubbiotti, A., Spadaccini, M., Badalamenti, M., Hassan, C., & Repici, A. (2022). Key factors for improving adenoma detection rate. Expert Review of Gastroenterology & Hepatology, 16(9), 819-833.

o   Ahmad, A., Wilson, A., Haycock, A., Humphries, A., Monahan, K., Suzuki, N., ... & Saunders, B. P. (2022). Evaluation of a real-time computer-aided polyp detection system during screening colonoscopy: AI-DETECT study. Endoscopy.

o   Troya, J., Fitting, D., Brand, M., Sudarevic, B., Kather, J. N., Meining, A., & Hann, A. (2022). The influence of computer-aided polyp detection systems on reaction time for polyp detection and eye gaze. Endoscopy, (AAM).

Brown, J.R.G., Mansour, N.M., Wang, P., Chuchuca, M.A., Minchenberg, S.B., Chandnani, M., Liu, L., Gross, S.A., Sengupta, N. and Berzin, T.M., 2022. Deep learning computer-aided polyp detection reduces adenoma miss rate: a United States multi-center randomized tandem colonoscopy study (CADeT-CS Trial). Clinical Gastroenterology and Hepatology, 20(7), pp.1499-1507.

Reply: According with the reviewer the references suggested have been included in the current version of the manuscript.

The authors need to explain more about Deskilling and its influence in different scenarios of polyps and respective changes in ADR due to deskilling.

Reply: A more extensive discussion regarding deskilling has been included in the new version of the manuscript:

“Hassan et al. showed that CADe detected polyps faster than the human eye but also had a high false positive rate. It was postulated that this could cause the examiner to become overly reliant on the AI and potentially result in a loss of endoscopic skills, especially for non-expert endoscopists (Hassan C, et al. Gut 2020;69:799-800). A recent experimental study used colonoscopy videos with and without CADe to assess the reaction time, visual gaze pattern, and the effect of false positive detections on the misinterpretation of mucosal surface as polyps. This study showed that with the use of CADe, eyes traveled a shorter distance, and endoscopists falsely identified more polyps, regardless of their degree of expertise. The authors suggested that overreliance on CADe could lead to a potential risk of deskilling (Troya J, et al. Endoscopy 2022;54:1009-14)”(Page 5 , lines 175-184).

-Additional text regarding deskilling and polyp characterization has been added (Page, Lines): “It has been suggested that these systems could decrease the learning curve in non-expert endoscopists. One study assessed whether a CNN trained for the characterization of colorectal polyps could increase the skills of endoscopists with different experience. The study found that the use of CNN led to an improvement in non-expert endoscopists (73.8 % to 85.6 %, P < 0.05) for colorectal polyp characterization, who almost reached the accuracy of experts (89.0% , P= 0.10). (Jin EH, et al. Gastroenterology 2020; 158:2169-79).” (Page 6 , lines 259-263, and Page 7, lines 264-265).

- Additional text has been added in the drawbacks section: “Deskilling, as mentioned above, is also a subject of concern. There is a risk of deskilling if endoscopists rely too heavily on the technology and do not continue to rely on their clinical skills and experience. Additional research is necessary to evaluate the impact of AI on the learning curve and to determine if the utilization of AI results in deskilling” (Page 9 , lines 365-366 and Page 10, line 367).

Round 2

Reviewer 1 Report

To the authors.

The mansucript has improved significantly.

I only still miss a discussion on the medico-legal and ethical aspects of the use of, and approval of using CADe and CADx in clinical practise. The in situ diagnostics seems to be incompatible with the current status where it can be used as supplement to analouge judgements only.

Author Response

We appreciate the comments of the Reviewer. We sincerely hope the answers  meet the expectations.

Reviewer 1

To the authors.

The manuscript has improved significantly.

I only still miss a discussion on the medico-legal and ethical aspects of the use of, and approval of using CADe and CADx in clinical practise. The in situ diagnostics seems to be incompatible with the current status where it can be used as supplement to analouge judgements only.

Reply: Thank you very much for this pertinent suggestion. The following paragraph has been included in the new version of the manuscript (page 9, lines 379-387): " Another area of improvement is the urgent need of a legal regulation for the use of AI in the healthcare setting[83]. Although, several CADe and CADx systems are currently commercialized, there are concerns that this new technology could be a source of healthcare mistakes, breaches in patient privacy and data security. The lack of transparency often comes up in legal discussions about the use of AI. Liability is also a source of concern. Many researchers emphasize the lack of understanding of the functioning of AI, as it is like a black box that produces predictions but cannot explain its results[84]. Although, there is currently no legal framework, some authors have suggested that the responsibility should be shared between the device constructors and the doctors who use it[83]."

Reviewer 2 Report

·       There is still not much discussion about the drawbacks and areas for improvement. Also, Section 3. Conclusions and future directions, still does not include enough and detail future direction. The following paper has a good example for these in Section V and VI.

Tavanapong W, Oh J, Riegler MA, Khaleel M, Mittal B, de Groen PC. Artificial Intelligence for Colonoscopy: Past, Present, and Future. IEEE J Biomed Health Inform. 2022 Aug;26(8):3950-3965. doi: 10.1109/JBHI.2022.3160098. Epub 2022 Aug 11. PMID: 35316197.

Author Response

We appreciate the comments of the Reviewer. We sincerely hope the answers meet the expectations.

Reviewer 2

There is still not much discussion about the drawbacks and areas for improvement. Also, Section 3. Conclusions and future directions, still does not include enough and detail future direction. The following paper has a good example for these in Section V and VI.

Tavanapong W, Oh J, Riegler MA, Khaleel M, Mittal B, de Groen PC. Artificial Intelligence for Colonoscopy: Past, Present, and Future. IEEE J Biomed Health Inform. 2022 Aug;26(8):3950-3965. doi: 10.1109/JBHI.2022.3160098. Epub 2022 Aug 11. PMID: 35316197.

Reply: We appreciate the comment of the reviewer. The following additional paragraph has been included in Section 3 ("Challenges, drawbacks and areas of improvement")(Please see Page 8, lines 352-371 and Page 9, lines 372-387): " There are concerns regarding the training requirements of this technology, clinical benefits, proper indications and cost-effectiveness. Some issues remain regarding training and transparency of CNNs[82]. One critical factor for training a CNN is the quantity and quality of the training dataset. It must be large enough and representative of the data obtained under real conditions. In addition, images should be adequately labeled. Ideally, the software should be able to adapt to different and suboptimal conditions. Obtaining a representative dataset may be difficult due to the many variables to consider, such as differences in colonic anatomy, cleansing quality, colonic normality variants, or endoscopist skills. In this scenario, class imbalance poses a challenge. This situation is a limitation for training a CNN under conditions of low prevalence. The prediction of the extent of invasion in early CRC would be an example. Methods such as data augmentation (an artificial procedure for increasing the dataset) consisting of random rotations, vertical and horizontal flips, and zoom can improve image classification. Additionally, active learning (an approach where an algorithm iteratively selects the most informative data samples to be labeled by a human expert) can make the CNN more robust. Another important issue is transparency. It is crucial to evaluate commercial AI systems available before the implementation in clinical practice. These systems probably have different algorithms, datasets, as well as their own training and validation. It is of utmost importance to have access to the image features that were used to train the CNN. Local interpretation methods explain why the network predicted a particular image, while global interpretation methods explain the entire model as a whole, including its capabilities and limitations. There must be transparency regarding both local and global explanations. This information could be valuable for hospital management and endoscopists when deciding on the eventual implementation of these devices in clinical practice. Another area of improvement is the urgent need of a legal regulation for the use of AI in the healthcare setting[83]. Although, several CADe and CADx systems are currently commercialized, there are concerns that this new technology could be a source of healthcare mistakes, breaches in patient privacy and data security. The lack of transparency often comes up in legal discussions about the use of AI. Liability is also a source of concern. Many researchers emphasize the lack of understanding of the functioning of AI, as it is like a black box that produces predictions but cannot explain its results[84]. Although, there is currently no legal framework, some authors have suggested that the responsibility should be shared between the device constructors and the doctors who use it[83].”

Reply: We have also included additional information in section 4 (Conclusions and future directions)(please see page 9, lines 411-425 and page 10, lines 426-435):" It should be noted that CADe has been shown to increase ADR, but at the expense of small and diminutive adenomas. Further research is needed to determine whether this translates into a reduction of CRC incidence and higher survival. Before the implementation of AI in clinical practice, some other issues require further research. First, the amount of colonic surface area examined is a crucial factor in determining colonoscopy quality. Although, AI devices only provide information of lesions within the field of view, they could potentially help to improve mucosal exposure. This, in turn, could increase lesion detection rates regardless of CADe devices. This real-time information could improve the quality of the examination. However, currently, there is low evidence regarding what information should be provided, how it should be presented, and where it should be shown. Initial studies with such systems demonstrated that the quality of colonoscopy could be similar to that performed by expert endoscopists[86]. Second, further efforts are needed to adapt AI tools to routine endoscopic technology. For instance, although recent studies have shown that AI-assisted polyp size measurement using laser technology can be more accurate than visual measurement, it requires special endoscopic equipment[87]. Another field of research is the integration of the AI throughout the colonoscopy process. The examination itself is just one part of the procedure, as endoscopists dedicate a significant amount of time after the colonoscopy to create the colonoscopy report, that includes quality measures such as withdrawal time and examined mucosal surface. AI can help to create structured reports and choose the most appropriate images, while also automatically rating cleansing quality based on validated cleansing scales. The integration of these AI systems in clinical practice can also provide individual and general data on the quality of endoscopists and endoscopy units, respectively. These data can be used for internal and external audits, which can help to meet quality standards and promote specific improvement strategies"

And also, this paragraph (Page 10, lines 447-450): "Finally, a field of development is robotics. There are currently self-propelling colonoscopes available on the market[89]. However, it is possible that in the future, colonoscopy and the different tasks performed during the procedure could be autonomously guided by software.”
